# Personalized Metabolic Profile by Synergic Use of NMR and HRMS

**DOI:** 10.3390/molecules26144167

**Published:** 2021-07-08

**Authors:** Greta Petrella, Camilla Montesano, Sara Lentini, Giorgia Ciufolini, Domitilla Vanni, Roberto Speziale, Andrea Salonia, Francesco Montorsi, Vincenzo Summa, Riccardo Vago, Laura Orsatti, Edith Monteagudo, Daniel Oscar Cicero

**Affiliations:** 1Department of Chemical Science and Technology, University of Rome “Tor Vergata”, 00133 Rome, Italy; petrella@scienze.uniroma2.it (G.P.); sara.lentini84@gmail.com (S.L.); ciufolini@scienze.uniroma2.it (G.C.); vanni@scienze.uniroma2.it (D.V.); 2Chemistry Department, University of Rome “Sapienza”, 00185 Rome, Italy; camilla.montesano@uniroma1.it; 3IRBM S.p.A., 00071 Pomezia, Italy; r.speziale@irbm.com (R.S.); vincenzo.summa@unina.it (V.S.); l.orsatti@irbm.com (L.O.); edith.monteagudo@chdifoundation.org (E.M.); 4Urological Research Institute, IRCCS Ospedale San Raffaele, 20132 Milan, Italy; salonia.andrea@unisr.it (A.S.); montorsi.francesco@unisr.it (F.M.); vago.riccardo@hsr.it (R.V.); 5Division of Experimental Oncology, URI Urological Research Institute, IRCCS San Raffaele Scientific Institute, 20132 Milan, Italy

**Keywords:** nuclear magnetic resonance, mass spectrometry, urine metabolome, normal ranges, personalized metabolic profile

## Abstract

A new strategy that takes advantage of the synergism between NMR and UHPLC–HRMS yields accurate concentrations of a high number of compounds in biofluids to delineate a personalized metabolic profile (SYNHMET). Metabolite identification and quantification by this method result in a higher accuracy compared to the use of the two techniques separately, even in urine, one of the most challenging biofluids to characterize due to its complexity and variability. We quantified a total of 165 metabolites in the urine of healthy subjects, patients with chronic cystitis, and patients with bladder cancer, with a minimum number of missing values. This result was achieved without the use of analytical standards and calibration curves. A patient’s personalized profile can be mapped out from the final dataset’s concentrations by comparing them with known normal ranges. This detailed picture has potential applications in clinical practice to monitor a patient’s health status and disease progression.

## 1. Introduction

Over the last hundred years, biochemical discoveries have made it increasingly possible to characterize the metabolic pathways in our bodies, develop new drugs, and monitor human nutrition and lifestyle. Though our knowledge is increasingly broad, it remains divided into specific areas, such as the characterization of genetic make-up or transcriptional factors underlying the expression of essential proteins involved in specific physiological or pathophysiological processes. In this context, metabolomics has come into its own to mend the cracks between the different disciplines hitherto used to study our biochemical mechanisms [1]. Considering that a single change in a DNA base can lead to the observation of alterations in metabolite concentrations of up to 10,000-fold changes [2], metabolomics represents a highly sensitive probe for depicting our phenotype. Helped by the development of new analytical technologies for obtaining and processing biochemistry data, metabolomics as an omics discipline is under constant development. In the last 20 years alone, more than 5000 papers have been published on the subject, making it one of the fastest-growing disciplines [3].

The most used analytical platforms in metabolomics are chromatography-mass spectrometry (LC–MS, GC–MS, CE–MS, and IMS–MS) and NMR spectroscopy. As reported in many papers, these two methodologies have several features that make them complementary [4]. For example, MS techniques are highly sensitive and allow for the detection of thousands of features at different concentration ranges, potentially expanding the description of a metabolic profile in detail with just a few microliters of sample. However, the identification of compounds by MS is a more complex process than by NMR. Indeed, the metabolite identity is solved by measuring the mass-to-charge ratio (*m*/*z*) of the ionized molecule and/or its ionized molecular fragments and then comparing them with reference spectra and/or using analytical standards [5]. Furthermore, not all MS techniques have the same degree of reproducibility; this is mainly the case for LC–MS measurements, which yield less reliable metabolite quantifications [6].

Unlike mass spectrometry, NMR is not a destructive technique and, in many cases, requires minimal sample preparation. The ability to determine the identity of a compound with a single analysis (^1^H-NMR) can be very accurate and fast for concentrated compounds or those that give signals in non-crowded regions of the spectrum. However, it has a lower sensitivity than MS [3], making it possible to only quantify a portion of the metabolome.

Thus far, few studies have described a combined use of both techniques, and these have directed their attention towards the development of statistical methods for weighing the two datasets [4] or for the structure determination of new compounds in commonly studied biofluids [7]. However, given the high complementarity of the two techniques, it should be beneficial to combine the data separately obtained with NMR and MS to improve the ability to classify and quantify the “metabotypes” under investigation [8].

One of the main limitations of using NMR is the relatively low number of accurately quantifiable metabolites in particularly complex mixtures like urine. For example, a study performed at two different fields (600 and 700 MHz) starting from a list of 151 metabolites that are potentially quantifiable in urine showed that only 50 presented data strongly correlated between the values obtained at the two magnetic fields [9]. This result represents a limit of quantifiable compounds in urine using NMR, and most studies in the literature have used a dataset of this size [10,11,12,13,14,15]. However, in one case, it was possible to reach 209 quantified metabolites [16], but only a fraction was detected in more than 80% of the samples.

Simultaneously identifying a metabolite by both NMR and MS would maximize the advantages for biomarker discovery by increasing the number of quantified metabolites in all samples and the accuracy of the measured concentrations. In our opinion, the method that has best combined MS data with NMR is the one developed by Nicholson et al. [17]. The authors named this strategy Statistical Heterospectroscopy (SHY) and showed that it is possible to correlate chemical shift and *m*/*z* data when a cohort of samples is considered. This concept revealed a new perspective to cross-reference NMR and MS data and to get the best of both techniques. However, the correlation was attempted with regions of the NMR spectrum, limiting the number of identifiable metabolites and obtaining only relative levels instead of concentrations. Our idea is to use the SHY concept to develop a novel strategy of the MS-assisted deconvolution of NMR spectra to extend the number of urinary metabolites quantified in their absolute rather than relative levels. We show how the synergistic use of both analytical methodologies can help to achieve this goal, taking the determination of metabolite concentrations in human urine as a specific case. We call this approach: *SY*nergic use of NMR and HRMS for *MET*abolomics (SYNHMET). Using SYNHMET, it was possible to obtain a complete dataset comprising 165 urinary metabolite concentrations for nine controls, six patients affected by chronic cystitis, and thirty-one bladder cancer patients.

## 2. Results

### 2.1. Metabolite Levels in Urine Acquired by NMR and HRMS

The SYNHMET method was first applied to quantify metabolites in urine samples from 46 subjects, divided into three groups: nine healthy controls (CTRL), six patients with chronic cystitis (CC), and thirty-one bladder cancer patients (BC). The ^1^H-NMR dataset was acquired using a 600 MHz Bruker Avance spectrometer (Bruker, MA, USA). The HRMS dataset was acquired using a UHPLC–high-resolution mass spectrometry (UHPLC–HRMS) analysis system coupled to an Orbitrap QExactive™ mass spectrometer (Thermo Scientific™, MA, USA) equipped with a HESI source operated in the positive and negative ion modes. In addition, we used two different chromatographic conditions: reverse-phase (RP), which allows for the separation of metabolites based on hydrophobic interactions, and hydrophilic interaction liquid chromatography (HILIC), enabling the analysis of polar compounds. Combining two ion modes and two chromatographic conditions allowed for broad coverage of urinary metabolites, an essential feature in an untargeted approach. After the MS analysis, 10,497 hits were obtained. Information about a matched formula, exact mass, retention time, and relative intensity was available for each hit, and in many cases, so was a putative name (Table 1).

### 2.2. Extraction of Urinary Metabolite Concentrations by SYNHMET

The proposed workflow for the SYHNMET method applied to the human urine samples described above is shown in Scheme 1. The following chapters illustrate its application focused on a specific region of the ^1^H-NMR spectrum.

#### 2.2.1. Creation of a Starting Profile and Deconvolution of NMR Spectra

Metabolites present in urine were first quantified by NMR using a deconvolution process. When this approach is applied, the goal is to minimize the difference between the experimental and calculated profiles. The latter is obtained by adding signals belonging to all the mixture components, weighted by their concentrations. To obtain a reasonable starting point for the first calculation, we selected 180 metabolites previously identified and quantified by NMR, considering their chemical shifts from the Chenomx database and their reported average concentrations in urine [16].

The SYNHMET procedure is illustrated using, as an example, the region between 2.47 and 2.37 ppm. The observed NMR profile of this zone mainly consists of the superimpositions of signals belonging to eleven metabolites: 2-oxoglutarate, 3-hydroxy-3-methylglutarate, 3-hydroxybutyrate, 4-pyridoxate, carnitine, glutamine, glutaric acid monomethyl ester, levulinate, pyroglutamate, succinate, and trans-4-hydroxy-L-proline.

In the first step, all chemical shifts and concentrations were changed to minimize the difference between calculated and experimental shapes. As a result, we obtained a list of metabolite concentrations constituted by very approximate values, especially those present at low levels or have their resonances hidden by other signals. The next step used MS-derived information to improve the accuracy of these measurements.

#### 2.2.2. Using the NMR/MS Correlation to Identify an MS peak(s)

The first step to incorporate the HRMS measurements involved creating a list for each metabolite containing all MS-detected peaks showing a difference lower than 5 ppm between their measured accurate masses and the monoisotopic molecular weight. After this search, all the eleven metabolites were linked to a variable number of MS-chromatographic peaks, ranging from seventeen (glutamine and 4-pyridoxate) to one (3-hydroxybutyrate). These numbers show the level of ambiguity in identifying an MS hit based only on the exact mass. We exploited the correlation between the MS intensities and the NMR concentrations obtained in the first step to assist MS identification.

Such an MS feature selection process is illustrated in Figure 1 for 2-oxoglutarate, which shows the correlations between NMR concentrations obtained in the first deconvolution round and five different chromatographic peak intensities. Despite the expected inaccuracy of these NMR concentrations, we could still identify the HC- peak at 4.57 min as 2-oxoglutarate (Figure 1e).

Using this procedure, we could unambiguously identify at least one MS feature for all other metabolites of this region, except for 3-hydroxybutyrate and trans-4-hydroxy-L-proline.

#### 2.2.3. HRMS Assisted NMR Deconvolution

The MS intensities of the assigned chromatographic peaks were converted into concentrations using the slope of the linear correlations. These values were then averaged with those measured by NMR, employed to update the profiles obtained after the first round, and finally applied these profiles as starting points for the second round. Finally, the second round of deconvolution was performed.

Figure 2a,b follows the evolution of the calculated profile for two samples after the first (upper panel) and second-round (lower panel) of deconvolution. Main variations are evident for low concentrated metabolites, such as 4-pyridoxate (blue) and 2-oxoglutarate (purple).

One thing worth noting is that there was no significant improvement in the agreement between the calculated (green) and experimental (black) profiles after the second round compared to the first one, although the metabolite concentrations on which they were based were very different. Because there were multiple ways to replicate the profile by combining the levels and positions of the metabolite signals, a precise reproduction of the measured spectrum did not warrant obtaining accurate concentrations. A deconvolution process assisted by the HRMS values would likely yield the most reliable results.

The ratios between first- and second-round concentrations were first calculated for each of the 46 samples. Next, the percent coefficients of variation (%CV) for each metabolite were calculated to assess the degree of change in concentration of this set of metabolites after mass-assisted deconvolution (Figure 2c). Metabolites showing low changes are the only ones that NMR can reliably quantify. On the other hand, the high percentage of variation for many metabolites shows the extent to which it is necessary to cross-reference the data between NMR and MS to obtain accurate results relying on two orthogonal measurements.

#### 2.2.4. Final Results

Table 2 summarizes the results obtained after two deconvolution rounds for the eleven metabolites clustered into the region of 2.47–2.37 ppm. Nine out of eleven metabolites could be unambiguously linked to one or more MS hits. Due to the different chromatographic conditions and polarities used, a single compound could be represented by more than one peak, such as for 4-pyridoxate, carnitine, and succinate. The final concentrations were calculated for all these nine metabolites by averaging the values obtained from the MS and NMR measurements, which showed a significantly improved correlation after the second round. This increase in the R^2^ value was a natural consequence of using the deconvolution process assisted by UHPLC–HRMS intensities, but its final value was still a measurement of the degree of agreement between these two datasets measured orthogonally. For 3-hydroxybutyrate and trans-4-hydroxy-L-proline, we were not able to identify any MS hits correlating with the NMR concentrations. In the first case, the final concentrations were still measured by NMR because we judged their values to be sufficiently reliable. On the contrary, the concentrations of trans-4-hydroxy-L-proline were not included in our final results. This compound has signals with high multiplicity, which divides the intensity into multiple components and does not present any resonance in a non-crowded region of the spectrum. These two facts make the NMR measurements inaccurate, hindering our ability to identify the correct MS hit(s) among the eleven showing its exact mass.

This procedure was repeated for all the regions into which the NMR spectrum was divided. As a result, we were able to quantify 165 metabolites out of the 180 initially considered. Of this total, twelve were quantified using only NMR data. Our final concentration matrix contained only 48 missing values, representing 0.6% of the total. These concentrations were normalized by converting them into μM/mM of creatinine, and they were compared with those of the literature (Appendix A). The excellent agreement between the retention times of nine labeled standards co-injected with the samples with those obtained by the HRMS-NMR correlation method (Appendix A) further sustains the assignments of the metabolites listed in Appendix A.

The set of metabolites quantified cover a wide range of biochemical markers, including amino acids and their metabolism, markers of vitamins, dysbiosis, diet and toxin exposure, carbohydrates and their metabolism, energy, fatty acid/lipid, and glycine/serine metabolism, and ketone bodies. In this way, it is possible to cover some of the central metabolic pathways for metabolomics studies to discover biomarkers related to pathological states and individual profiling.

### 2.3. The Origins of the Analytical Synergism between NMR and UHPLC–HRMS

Many metabolites are difficult to quantify by NMR, mainly those in low concentrations and those whose signals are hidden by the presence of other resonances. We have defined this interference as the “NMR matrix effect” because its consequence is the same as observed when ESI is used as the ionization source [18]. Its impact differs from one spectrum to another, mainly in biofluids like urine that show highly variable composition. The use of MS intensities characterized by an exact mass and measured after a chromatographical separation highly alleviates this difficulty because the quantification is performed on separated components of the mixture. Combining the NMR and HRMS datasets offers the opportunity to obtain accurate concentrations for many metabolites, a result that would be impossible to achieve when using NMR alone.

However, if, on the one hand, the use of UHPLC–HRMS data expands the number of quantified metabolites, on the other hand, NMR aids in increasing the accuracy of MS-derived concentrations. The most frequent causes of error in the evaluation of concentrations by MS are the detector’s saturation and the matrix effect. These two effects are unpractical to correct when a large matrix composed of a considerable number of samples and metabolites is analyzed in an untargeted MS-based analysis.

An example of the first case was found during the quantification of hippuric acid, which shows a wide range of concentrations in urine [19,20]. Figure 3 shows that the response of the MS detector was not linear for concentrations above 3.8 mM. Thus, we only considered NMR values for those samples with values above this limit to avoid significant errors.

The second cause leading to MS quantification errors may be even more challenging to detect. An example was found in one sample of a BC patient, for which Table 3 shows the different hippuric acid concentrations calculated from the MS intensities in the four conditions and NMR. The concentrations derived from HC chromatography agreed with the NMR data, while those measured with the RP column were significantly lower. In this case, only the values obtained with the first chromatographic condition were considered. This effect was not detected in other samples and was probably due to a compound present only in this case that co-elutes with this metabolite, causing a partial suppression of the peak intensity.

In conclusion, the synergy between these two techniques is reflected because MS mainly contributes to quantifying a metabolite in those samples where it is present at low levels or with hidden signals. At the same time, NMR does so for those showing a higher concentration or isolated signals, thus providing the key to identify the different chromatographic peaks and correct errors in the MS dataset due to saturation or matrix effect.

### 2.4. Personalized Metabolic Profile from SYNHMET Application

Metabolite concentrations need to be normalized to account for the variable hydration status of a subject before assessing the normality of their values. Routinely, this normalization is performed by the creatinine level [21,22,23,24]. Its concentration is also a criterion for selecting or rejecting the sample for metabolic profiling. According to the World Health Organization (WHO), only urine samples with creatinine concentrations in the range of 0.3–3.0 g/L are acceptable [25]. One sample of our set was discarded for this reason; it had a low creatinine level (0.15 g/L).

Subsequently, we compared the normalized concentrations for all the other subjects with the normal ranges reported for adults over 18 years of age (Figure 4). Almost all concentration values for the CTRL group fell within the normal ranges. Only one CTRL subject showed higher than normal values for threonine and carnosine concentrations. On the contrary, the profiles from the CC and BC groups showed a much higher number of metabolites with abnormal values. Those indicated with black in Figure 4 were more than four times higher than the maximum literature value. These anomalies most likely reflected different metabolic imbalances related to the pathologies of these patients.

Specifically, for the BC group, 82 values were found to lie outside the literature ranges. Most abnormal values corresponded to dietary components, followed by metabolites belonging to fatty acids/lipids, carbohydrates, energy, and branched-chain amino acid metabolisms. Nine metabolites previously found significantly altered in BC patients—namely O-acetylcarnitine, gluconate, lactate, phenylacetylglutamine, citrate, hippurate, succinate, valine, and erythritol [26]—were also found outside their normal ranges (Figure 4). The complete metabolic profile of one BC patient is shown in Appendix A. Twenty-four metabolic concentrations lay outside the literature ranges (Figure 5). They primarily belonged to components of the diet, fatty acid metabolism, and energy metabolism. These results underline the degree of detail that can be achieved with the SYNHMET methodology, with a potential clinical practice application to monitor apatient’s health status and disease progression.

## 3. Discussion

The value of combining the two most commonly used techniques in metabolomics, NMR and MS, was recently recognized and addressed in a review by Marshall and Powers [8]. However, no method has attempted to directly correlate an NMR chemical shift with an MS *m*/*z* value of a single sample because there is no specific information to indicate that these two features belong to the same molecule [8]. MS intensities belonging to a cohort of samples were cross-correlated with NMR spectral regions to overcome this limitation [17]. The correlation that does not exist in one sample exists in all of them as a group because, at this point, it is the distribution of intensities that determines whether a given chemical shift belongs to a molecule signal presenting a certain *m*/*z*. The so-called Statistical Heterospectroscopy approach, however, led to the identification of a reduced number of metabolites. The major drawback of this approach is, in our opinion, that the correlation was attempted between intensities of compounds whose levels are measured separately by HPLC–MS with regions of the NMR spectrum whose intensities result from the simultaneous contributions of many metabolites. A clear correlation between the MS and the NMR bin intensities can only be expected for strongly dominating metabolites because of their concentration in the region’s shape.

Differently, SYNHMET uses the resolution power of NMR to separate most of the different signals contributing to the spectrum profile, coupled to that of UHPLC–HRMS. The deconvolution strategy was used to extract more than 200 metabolite concentrations from urine [16], a result not reproduced in any further study to the best of our knowledge. The difficulty associated with this methodology lies mainly in the extraction of levels for not concentrated metabolites or those presenting signals in crowded regions. These areas only provide the sum of the contributions of the various compounds, and without further information, there are many ways to combine the positions and intensities of the mixture components to reproduce the experimental shape of the NMR spectrum. The simultaneous use of UHPLC–HRMS intensities provides the key to obtain a single solution because it adds two new features to calculate the relative contribution of metabolites to a profile: the molecular weight and the chromatographic resolution. The latter is not practical in NMR measurements due to a combination of low sensitivity and long acquisition times. In this way, the correct proportions are extracted by combining NMR and UHPLC–HRMS, which transforms the experiment used for metabolite identification/quantification from monodimensional (chemical shift) into three-dimensional (by adding retention time and exact mass). Globally, the mechanism by which this method operates can be defined as an MS-assisted NMR deconvolution, improving the quality and quantity of the obtained data compared to that expected when exclusively using NMR (Scheme 2).

According to the Metabolomics Standard Initiative, a definite metabolite identification, called level 1, needs a direct comparison of experimental data with an authentic reference standard [27]. It was argued that NMR metabolite identification of compounds in mixtures achieved by comparing with a spectra database approaches level 1 identification [27]. In the SYNHMET strategy, we added parameters characterizing a compound (chemical shift, multiplicity, and the number of signals) and the elemental composition provided by the correlation with the MS data to the NMR. This additional information limits the possible structures to the existing isomers, constituting a very restricted chemical space for low molecular weight compounds. The probability that two isomers show the same NMR parameters is extremely low, if even possible. For all these reasons, the confidence in the SYNHMET identification should be considered, in our opinion, similar to that in level 1.

Applying SYNHMET enabled us to quantify a large number of metabolites in urine. Many papers have supported the concept that the utility of a given approach is directly proportional to the measurable number of metabolite levels. However, this is only one of the two essential parameters in defining the value of a dataset for metabolomics studies. The other is the completeness of the matrix because if there are too many missing values, the classification ability or the detection of correlations between metabolites becomes weaker [28]. In our experience and from analyzing the literature on NMR urine metabolomics, the maximum number of metabolites quantified in at least 80% of samples is around 50–60 [10,11,12,13,14,15]. This number is far from that achieved in LC-MS studies, which have reached more than a thousand [29]. However, the identity of many of these metabolites is only putative because it is only supported by fragmentation spectra.

In our scheme, both the reproducibility and accuracy of the results are mainly supported by the characteristics of NMR. It is commonly accepted that these are two main robust features of NMR, which involve the possibility of obtaining the same instrumental response even when different spectrometers are used. These characteristics have favored constructing a community-built reference calibration line, with the participation of twenty-three laboratories, including ours [30]. We foresee that a similar calibration line can be produced among laboratories keen to prove the validity of the SYNHMET approach, allowing for the scientific community to obtain more robust results in metabolomics. However, the number of samples analyzed in this work did not allow for a definitive answer, and further studies will be needed to assess the limits in terms of precision and accuracy.

A compelling application of SYNHMET is the possibility of generating a detailed personalized profile of urinary metabolites. The main way to get a reliable profile is the election of an effective way to normalize the metabolites’ concentrations to correct the variation induced by the subject hydration status. Usually, concentrations are normalized by the total urine volume collected during 24 h or the urinary creatinine level. These two normalization strategies present advantages and drawbacks. In case of using the total urine volume for 24 h, the incomplete collection is the main problem [22]. On the other hand, creatinine concentration is affected by several factors that are not directly related to the glomerular filtration rate, like muscle mass, diet, age, sex, and race [31,32]. Creatinine is also secreted from the renal tubules, which is not desirable for a glomerular filtration marker. A study comparing the uncertainties related to standardization of urine samples with volume and creatinine concentration showed that the latter introduces a 19–35% error [22]. However, if compared with the total volume normalization, this is partially counteracted by the higher risk that the sample is incomplete in collecting voids during a 24-h time interval. More recently, a study showed that normalization with urinary creatinine is better than volume in rats under controlled preclinical conditions, even when compared to a more recently proposed normalizer, cystatin C [33].

Our study used creatinine to normalize the metabolite concentrations, mainly because almost all the available normal ranges found in the literature are expressed in μM/mM of creatinine [34]. Independently on the used strategy, the profile of normalized metabolite concentrations constitutes a personalized urinary picture, which can be used to expand the current capability of classical biochemical tests to determine aperson’s health status. This approach is very different from classical metabolomics, which seeks to find universal biomarkers of a disease or drug effects. The concept of personalized medicine grew up from the scientific evidence that there is high interindividual variability in the metabolic response to any change in the health status or the response to a drug. Therefore, expanding the number of metabolites that can be routinely monitored in biofluids can define a more accurate picture to be used in clinical practice [35]. At the heart of this analysis is the concept that a person’s metabolic profile can reflect an individual’s overall health status. Nowadays, physicians only capture a tiny fraction of the information contained in the metabolome, mainly due to its high complexity and the lack of robust and efficient analytical methods to determine the absolute instead of the relative level of a large number of chemical compounds in biofluids. Routine analyses only evaluate a very restricted number of compounds, such as glucose level for monitoring diabetes, cholesterol and low/high-density lipoproteins for cardiovascular health, or urea and creatinine for renal disorders. Simultaneously determining the absolute concentration of hundreds of molecules will open up new scenarios towards more accurate personalized medicine and increase the predictive value of such analyses.

For example, a patient suffering from BC showed a urinary profile with significant abnormal values for metabolites belonging to galactose/starch sucrose, caffeine, and lysine metabolisms (Figure 5). A recent study about recognizing different stages of BC using machine learning identified the first two as the main dysregulated metabolisms in early stages, whereas lysine metabolism was found to be unbalanced in late stages [36]. The case of caffeine metabolism is remarkable. Along with one of its metabolites, 1,3-dimethylurate, caffeine is processed by a P450 family cytochrome acting in the liver, CYP1A2 [37]. The connection between caffeine metabolism, exposure to tobacco compounds, and urinary mutagenicity has been known for a long time [38]. Significantly, cigarette smoking is the leading risk factor for BC, accounting for 50% of the total [39]. In addition to this patient, urine caffeine levels were significantly elevated in six other subjects with BC.

This patient also presented significant comorbidity due to cardiovascular pathologies, particularly severe myocardial ischemia. We observed different altered metabolisms related to cardiopathies, like those corresponding to branched-chain amino acids, lactate, and fatty acid metabolism [40]. They are the consequences of increased fatty acid metabolism, decreased glucose metabolism, and impaired branched-chain amino acid catabolism. Finally, the patient showed chronic pancreatitis, probably related to past alcohol abuse. The malfunction of the pancreas should explain the very high level of glucose in the urine, as in diabetic subjects.

## 4. Materials and Methods

### 4.1. Chemicals and Reagents

All used solvents and reagents were LC–MS grade. Water (H_2_O), acetonitrile (ACN), formic acid (FA), and ammonium formate (CAS 540-69-2) were obtained from Sigma Aldrich (St. Louis, MO, USA). The stable isotope-labeled (SIL) internal standard ^13^C^15^N_2_-8-hydroxy-2′-deoxyguanosine (^13^C^15^N_2_-8-OH-dG) was obtained from Toronto Research Chemicals (Toronto, ON, Canada). ^15^N_4_-hypoxanthine (^15^N_4_-Hyp), L-tyrosine-(phenyl-d_4_) (d_4_-L-Tyr), and ^15^N_4_-inosine (^15^N_4_-I) were purchased from Cambridge Isotope Laboratories, Inc., (Tewksbury, MA, USA). L-kynurenine sulfate: H_2_O (ring-d_4_, 3,3-d_2_) (d_6_-KYN) and D_2_O were acquired from Cambridge Isotope Laboratories, Inc. (Andover, MA, USA). Anthranilic acid-ring-^13^C_6_ (^13^C_6_-AA) and 3-(trimethylsilyl)-2,2,3,3-d propionic acid (TSP) were purchased from Sigma Aldrich (Schnelldorf, Germany). ^15^N,^13^C_2_-3-Hydroxy-DL-kynurenine (^15^N-^13^C_2_-OH-KYN) was obtained from AMRI (Albany, NY, USA).

### 4.2. Urine Collection

Urine samples were obtained from the Urological Research Institute (URI) of San Raffaele Hospital (Milan, Italy). Caucasian patients aged between 32 and 90 years were recruited. The dataset comprised 46 samples: 31 bladder cancer (BC) patients, nine healthy controls, and six with chronic cystitis. BC patients with concomitant or previous prostate, renal, or upper excretory tract cancer; urinary tract infections; or kidney failure were excluded. Urine samples were collected before the surgical intervention and processed soon after. The samples were centrifuged at 300 g for 5 min, aliquoted, and stored at –80 °C until use.

### 4.3. UHPLC–High Resolution Mass Spectrometry Analysis

#### 4.3.1. SIL-Stock and Working Solution Preparation

Stock solutions were prepared from the independent weight of compounds and stored at −20 °C. d_6_-KYN and ^13^C_6_-AA were prepared in H_2_O/DMSO (1/1, *v*/*v*) at 1.5 and 5.0 mg/mL, respectively. ^15^N-^13^C_2_-OH-KYN was prepared in H_2_O/DMSO (1/19, *v*/*v*) at 2.0 mg/mL. ^13^C^15^N_2_-8-OH-dG, ^15^N_4_-Hyp, d_4_-L-Tyr, and ^15^N_4_-I were prepared in water at 1.0 mg/mL.

Internal Standard Working Solutions (IS-WS) were prepared by adding appropriate volumes of the stock solutions to 50 mL of ultrapure H_2_O (ISWS-A) and ACN (ISWS-B) to reach a final concentration of 200 ng/mL for all the standards. The solutions were maintained at 4 °C and freshly prepared every week.

#### 4.3.2. Urine Normalization by Specific Gravity

Specific gravity (SG) measurements were made with a portable digital refractometer (Atago UG-α, Tokyo, Japan). The refractometer had a urinary SG range from 1.000 to 1.060 with a resolution of 0.001. Urine samples were thawed at room temperature in an ultrasonic bath for 10 min and then centrifuged (4000 rpm). An aliquot of urine (100 µL) was placed upon the lens of the refractometer previously calibrated with LC–MS-grade water to measure SG values. Samples were then split into two aliquots. Urinary metabolite levels were normalized by SG-diluting each aliquot with water or ACN:H_2_O in variable amounts for RP and HILIC analysis, respectively. Dilutions were performed to bring all samples to the same specific gravity value.

#### 4.3.3. Urine Samples Preparation

All samples were further diluted by 3-fold with ISWS-A for RP analysis or ISWS-B for HILIC analysis. Samples were vortexed and centrifuged (13,000 g for 10 min), and the supernatant (350 μL) was transferred to a 96-well plate and randomized for LC–MS analyses.

#### 4.3.4. Quality Control Samples and Blanks Preparation

Two different types of quality control (QC) samples were prepared: pooled QCs made by mixing equal volumes (5 μL) from each sample previously normalized for the specific gravity and dilution QCs prepared by 2, 4, and 8-fold diluting the pooled QCs with LC–MS-grade water. All QCs were further diluted by 3-fold with ISWS-A for RP analysis or ISWS-B for HILIC analysis. Pooled QC samples were injected first (*n* = 20) to condition the LC–MS system and obtain stable retention times and MS response. Subsequently, pooled QCs were injected every six true samples (*n* = 8 in total) to perform intra-batch signal drift corrections. Dilution QCs were analyzed four times and were regularly incorporated along the sample list to verify the linear response of the MS signal. Blanks consisted of LC–MS-grade water for RP analysis and ACN: H_2_O 80:20 (*v*/*v*) for HILIC analysis. Blank injection (*n* = 3) was performed at the beginning of the batch to collect a background signal excluded from the dataset.

#### 4.3.5. HILIC and RP Chromatography

The used UHPLC system was an Ultimate 3000™ liquid chromatographic system (Thermo Scientific™, MA, USA) coupled to an Orbitrap Q Exactive™ mass spectrometer (Thermo Scientific™, MA, USA) equipped with a HESI source operating in the positive and negative ion modes. HILIC chromatographic separation was accomplished using a BEH-HILIC column, 130 Å, 1.7 μm, and 2.1 × 100 mm (Waters, Milford, MA, USA). The used mobile phases were: 20 mM ammonium formate along with 0.1% FA at pH 3.7 (mobile phase A) and ACN (mobile phase B). The gradient consisted of a linear increase of mobile phase B from 5% to 35% over 8.5 min, followed by an additional increase to 50% in 1 min. Phase B was kept constant for 1.5 min and then decreased to 5% in 0.5 min and kept stable for 3.5 min for column re-equilibration (total run time of 15 min). The used flow rate was 0.300 mL/min, the injection volume was 2 µL, and the column was kept at 35 °C.

RP chromatographic separation was achieved using an HSS-T3 column, 100 Å, 1.7 μm, and 2.1 × 100 mm (Waters, Milford, MA, USA). The mobile phases were: 0.1% FA in H_2_O (mobile phase A) and 0.1% FA in ACN (mobile phase B). The gradient ramp consisted of a linear increase to 10% of mobile phase B over 6 min and to 35% in 2 min. Mobile phase B was further increased to 98% in 2 min, kept constant for 0.5 min, and finally decreased to 0% in 0.5 min and kept stable for 3 min for column re-equilibration (total run time of 15 min). The flow rate was 0.300 mL/min from 0 to 8.0 min, increased to 0.4 mL/min from 8.0 to 12.0 min for column washing, and brought back to 0.3 mL/min from 12.0 to 15.0 min. The injection volume was 2 µL, and the column was kept at 35 °C. During LC–MS analysis, samples were kept in the autosampler at 8 °C.

#### 4.3.6. High-Resolution Mass Spectrometry

Mass spectra were acquired on an Orbitrap QExactive™ mass spectrometer (Thermo Scientific™, MA, USA) operating in both the positive and negative ion modes. The HESI parameters were: 3.20 kV (pos)/–3.20 kV (neg) electrospray voltage, 280 °C heated capillary temperature, 50 (pos)/–50 (neg) S-lens RF level, sheath gas (N_2_) flow of 50 a.u., auxiliary gas (N_2_) flow of 10 a.u., and gas temperature of 300 °C. The acquisition range was set from *m/z* 60 to 900 at a resolution of 70,000 FWHM at *m/z* 200. All data were acquired in profile mode using Xcalibur™ 3.1.66.10. The QExactive™ mass spectrometer was calibrated for the positive and negative modes before sample analysis using the calibration solution provided by the manufacturer (Pierce LTQ ESI Positive Calibration Solution and Pierce LTQ ESI Negative Calibration Solution). For the mass calibration of the instrument, a custom list that included lower masses than the default calibration provided with the instrument was used to ensure that accurate masses were detected at low molecular weights.

#### 4.3.7. Raw Data Processing by Compound Discoverer

The raw files obtained in the positive and negative ion modes were separately processed using Compound Discoverer™ 2.0 (Thermo Scientific™). Four output tables (RP+, RP-, HILIC+, and HILIC-) were generated, including *m/z*, retention time, and peak intensity, for all the analyzed samples. An untargeted metabolomics workflow for retention time alignment, component detection, elemental composition prediction, and gap-filling was used. The workflow tree included the following nodes: input files, select spectra, align retention times, detect unknown compounds, group unknown compounds, fill gaps, normalization areas, and mark background compounds. The raw files were aligned with an adaptive curve setting with a 5 ppm mass tolerance and a 0.4 min retention time shift. Unknown compounds were detected with a 5 ppm mass tolerance, signal to noise ratio of 3, 30% of relative intensity tolerance for isotope search, and 10,000 minimum peak intensity, and then they were grouped with 5 ppm mass and 0.3 min retention time tolerances. A procedural blank sample was used for background subtraction and noise removal during the pre-processing step. Peaks were removed from the list if they showed less than a 3-fold increase compared to blank samples or if they were detected in less than 50% of QCs and/or with relative standard deviation (%RSD) of the QCs greater than 50%. To balance differences in intensities that may have arisen from instrument instability, a normalized area across all samples was provided for each detected metabolic feature by normalization to the periodically analyzed QC samples (pooled QC).

Finally, the hit intensities of each sample were multiplied by the dilution factor used for pre-normalization. Thus, un-normalized data were used to ensure a better degree of correlation between NMR and MS.

### 4.4. H-NMR Spectroscopy

#### 4.4.1. Sample Preparation

The urine samples, previously stored at –80° C, were thawed on ice and centrifuged at 4000 rpm for 10 min at 4 °C; then, 500 µL of supernatant were collected. Then, 50 µL of phosphate buffer solution [41] (1.5 M K_2_HPO_4_/NaH_2_PO_4_, 30 mM NaN_3_, and 5.5 mM TSP, pH 7.4 in D_2_O) were added, and 50 µL of the final solution were transferred to a 1.7 mm thin-walled glass NMR tube for subsequent NMR analysis.

#### 4.4.2. Spectra Acquisition

^1^H-NMR experiments were performed on Bruker Avance 600 MHz equipped with a SampleJet autosampler using a noesypr1d sequence, mixing time of 100 ms, a spectral window of 12 ppm, acquisition time of 2 s, relaxing time of 3 s, 516 scans, 4 dummy scans, and T = 298 K. This sequence has become the best choice for NMR-based metabolomics studies [42] for several reasons. Firstly, the quality of water suppression is very high without the need for extensive optimization. Secondly, an increasing number of well-established groups utilize the sequence, reflecting its consistency [43]. Finally, the library of Chenomx used in this study to quantify metabolite concentrations is optimized for this sequence and compensates for incomplete relaxation.

#### 4.4.3. H-NMR Data Analysis

All the spectra were processed using 0.5 Hz of line-broadening followed by manual phase and baseline correction. Chenomx NMRSuite 8.5 (Chenomx Inc.) was used to quantify the concentrations of the metabolites. The spectra database in this software allows for the manual deconvolution of different signals and determines the concentration of the compounds that form the mixture. TSP was set as an internal standard at 0.5 mM.

### 4.5. SYNHMET Method

The starting spectrum profile for deconvolution is defined using the average concentrations of urine metabolites [16]. The chemical shifts and levels of all compounds are then varied to reproduce the profile observed in each experimental NMR spectrum. The matching between the calculated and experimental spectral profiles is never perfect. The source of this inequality can be understood by analyzing all variables contributing to the spectrum intensity at a given chemical shift (*I_k_*) (Equation (1)):(1)$$I_{k}\;=\;\overset{n}{\underset{i\;=\;1}{\sum }}a_{i}K_{i,k}+\;\overset{m}{\underset{j\;=\;1}{\sum }}b_{j}U_{j,k}+N_{k}$$
where *k* is the chemical shift, *i* represents one assigned metabolite, *n* is the total number of assigned metabolites, *K_i,k_* is a known factor accounting for the shape of assigned metabolites, *j* represents one unassigned metabolite, *m* is the total number of unassigned metabolites, *U_j,k_* is an unknown factor considering the shape of unidentified metabolites, *a_i_* and *b_j_* are the metabolite concentrations, and *N_k_* is a random factor representing the noise.

In parallel, the exact mass of each metabolite is searched in the MS dataset, creating a list of linked MS features for most compounds. The number of MS peaks associated with each metabolite varies from zero to more than twenty. Detecting more than one peak with the same exact mass turns the identification based solely on the molecular weight uncertain unless using labeled standards. In the SYNHMET method, combining the concentrations measured for a cohort of samples simultaneously by MS and NMR can solve this ambiguity in an alternative way. We considered that a certain MS-detected chromatographic peak showing the accurate mass of a metabolite can be attributed to it when it is the only one showing a significant correlation between the distributions of the MS peak intensities and NMR concentrations. The intensities of the selected peak are then converted into concentrations by multiplying them with the slope of the best fit solution. The initial spectrum profile is then adjusted, inserting the values of the peak or peaks averaged to those measured by NMR for each metabolite. Conversely, concentrations of compounds not represented by any MS feature or showing multiple or no correlations are not updated for the following phase.

During the next profiling step, all compounds’ signal positions and concentrations defining the updated profile are varied to obtain the best accordance between the calculated and experimental profiles. After completion, a new correlation test is accomplished, possibly increasing the number of identified and consequently quantified metabolites. This process is iteratively repeated until no further information is added. The final matrix contains concentrations of metabolites that are determined by a combination of MS and NMR measurements.

## 5. Conclusions

In conclusion, the new methodology for merging NMR and UHPLC–HRMS produced a list of 165 metabolite concentrations in urine in almost all samples, with significantly higher accuracy of identification and quantification than could be reached separately using the two techniques. In addition, its application allowed us to delineate a personalized urinary profile based on a list of compound levels covering a wide range of metabolic processes. Its expansion to more samples in the future will allow us to enlarge our knowledge of many metabolites’ normal and abnormal values in human urine. Its translation into clinical practice can be of great value, such as identifying biomarkers of disease susceptibility and following the individual therapeutic outcomes [35]. These two aspects are among the main applications of metabolomics to improve the accuracy of personalized medicine.

## Data Availability

Not applicable.

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
