# Peer review of "Personalized Metabolic Profile by Synergic Use of NMR and HRMS"

_molecules, 2021, doi:10.3390/molecules26144167_

Round 1

Reviewer 1 Report

1) This manuscript describes a new strategy to identify and quantify metabolites in urine by combining NMR, UHPLC, and HRMS techniques. This methodology, called SYNHMET, is based on iterative correlation between quantitative NMR and MS analyses. Coupled HILIC and RP chromatography to HR-mass spectrometry determines exact masses of metabolites, but their identification and most importantly absolute quantification are achieved from NMR spectra, although poorly resolved. Decomposition of the NMR spectra is performed basing on the best NMR-MS signal intensity correlation. SYNHMET is applied to analysis 46 samples of healthy controls as well as chronic cystitis and bladder cancer patients, providing a wide range of metabolic profiles. I think the result is interesting enough for publication in “molecule” after careful revision.

2)  typos: lines 21, 239, 478, 576.

3) Line 101: Bruker is a German company, it is better to change to (Bruker, Karlsruhe, Germany).

4)  Figure 2:

a) Please mention in caption which line corresponds to the final model spectrum, i.e., the sum of calculated components. I guess the green line.

b) The variations shown in c) correspond to which sample? Or probably to all the 46 samples and the averages are presented. Please precise.

5)  Table S1: Please check footnote for e) and f): which references?

6)  Figure 5:

a) Please indicate the unit for the values given.

b) It is not clear what the color codes represent. Are they statistics of the 46 samples studied? The same for Figure S1.

7)  Line 575: Why Noesy-1d sequence is used, particularly when we want quantitative NMR?

8)  Reference section: Lines 690 and 710: journal names for ref #22 and 31 have to be indicated.

Author Response

1) This manuscript describes a new strategy to identify and quantify metabolites in urine by combining NMR, UHPLC, and HRMS techniques. This methodology, called SYNHMET, is based on iterative correlation between quantitative NMR and MS analyses. Coupled HILIC and RP chromatography to HR-mass spectrometry determines exact masses of metabolites, but their identification and most importantly absolute quantification are achieved from NMR spectra, although poorly resolved. Decomposition of the NMR spectra is performed basing on the best NMR-MS signal intensity correlation. SYNHMET is applied to analysis 46 samples of healthy controls as well as chronic cystitis and bladder cancer patients, providing a wide range of metabolic profiles. I think the result is interesting enough for publication in "molecule" after careful revision.

2)  typos: lines 21, 239, 478, 576.

All typos have been corrected.

3) Line 101: Bruker is a German company, it is better to change to (Bruker, Karlsruhe, Germany).

Bruker was a German company, now is an American company with headquarters at Billerica, Massachusetts, US.

4)  Figure 2:

  1. a) Please mention in caption which line corresponds to the final model spectrum, i.e., the sum of calculated components. I guess the green line.

We modified Figure 2's caption by adding the following sentence (line 194): "Black and green lines represent the experimental and the calculated spectrum, respectively."

  1. b) The variations shown in c) correspond to which sample? Or probably to all the 46 samples and the averages are presented. Please precise.

The following paragraph was added at line 182 to explain this point:

"The ratios between first- and second-round concentrations were first calculated for each of the 46 samples. Next, the percent coefficients of variation (%CV) for each metabolite were calculated to assess the degree of change in concentration of this set of metabolites after mass-assisted deconvolution (Figure 2(c))."

5)  Table S1: Please check footnote for e) and f): which references?

The corrected Table S1 has only three footnotes: The reference for footnote c) is now indicated.

6)  Figure 5:

  1. a) Please indicate the unit for the values given.
  2. b) It is not clear what the color codes represent. Are they statistics of the 46 samples studied? The same for Figure S1.

The caption of Figure 5 now states: " Urinary metabolites of a BC patient showing abnormal values according to literature ranges. Blue and red areas represent 10% lower and higher values than those reported in the literature for adults over 18 years old, respectively. All values are expressed in mM/mM of Creatinine."

7)  Line 575: Why Noesy-1d sequence is used, particularly when we want quantitative NMR?

Now we have explained why we use the sequence at line 654:

" This sequence has become the best choice for NMR-based metabolomics studies [41], based on several reasons. Firstly, the quality of water suppression is very high without the need for extensive optimization. Secondly, an increasing number of well-established groups utilize the sequence, reflecting its consistency [42]. Finally, the library of Chenomx used in this study to quantify metabolite concentrations is optimized for this sequence and compensates for incomplete relaxation."

Accordingly, two new references were added.

8)  Reference section: Lines 690 and 710: journal names for ref #22 and 31 have to be indicated.

Journal names have been inserted for ref #22 and #31:

22) Miller, R.C.; Brindle, E.; Holman, D.J.; Shofer, J.; Klein, N.A.; Soules, M.R.; O'Connor, K.A. Comparison of Specific Gravity and Creatinine for Normalizing Urinary Reproductive Hormone Concentrations. Clin. Chem. 2004, 50, 924–932, doi:10.1373/clinchem.2004.032292.

31) Adedeji, A.O.; Pourmohamad, T.; Chen, Y.; Burkey, J.; Betts, C.J.; Bickerton, S.J.; Sonee, M.; McDuffie, J.E. Investigating the Value of Urine Volume, Creatinine, and Cystatin C for Urinary Biomarkers Normalization for Drug Development Studies. Int. J. Toxicol. 2019, 38, 12–22, doi:10.1177/1091581818819791.

Reviewer 2 Report

in file

Author Response

The authors of the manuscript present an interesting concept of using the synergy of two complementary methods (proton NMR spectroscopy and HRMS mass spectroscopy) to improve the quantification of concentrations of a large number of compounds in biofluids. The new strategy allows for much better results than using both techniques separately. The authors demonstrate the effectiveness of the method by measuring the urine of healthy people and patients with chronic cystitis and bladder cancer.

The authors report that they achieved this result. This result was achieved without the use of analytical standards and calibration curves. In addition, the personalized patient profile can be mapped from the concentrations of the final data set by comparing them to known normal ranges. The authors conclude that this detailed picture has potential applications in clinical practice for monitoring patient health and disease progression.

Here, there are two issues to be resolved:

  1. Whether the lack of calibration gives a real possibility of repeatability of results. The sample for analytical research was not too

An answer based on the article is not really possible. Too few samples tested and the existence of certain ambiguities.

What can be improved?

A procedure should be described in the context of verifying the accuracy of the proposed complementary NMR approach over HRMS.

We have addressed these points in the discussion by adding the following paragraph at line 404:

" In our scheme, both reproducibility and accuracy of the results are mainly supported by the characteristics of NMR. It is commonly accepted that these are two main robust features of NMR, which involve the possibility of obtaining the same instrumental response even when different spectrometers are used. These characteristics favored constructing a community-built reference calibration line, with the participation of twenty-three laboratories, including ours [29]. We foresee that a similar calibration line can be produced among laboratories keen to prove the SYNHMET approach, allowing the scientific community to obtain more robust results in metabolomics. However, the number of samples analyzed in this work does not allow a definitive answer, and further studies will be needed to assess the limits in terms of precision and accuracy."

Accordingly, we have added a new reference.

  1. Can we really talk about the attractiveness of this approach from the point of view of applications and requirements of clinical trials, as the authors suggest?

In the text we did not mention clinical trials, but clinical practice. To further clarify what is in our opinion the value of this approach in clinical analysis, we have added the following paragraph at line 471:

" At the heart of this analysis is the concept that a person's metabolic profile can reflect the individual's overall health status. Nowadays, physicians capture only a tiny fraction of the information contained in the metabolome, mainly due to its high complexity and the lack of robust and efficient analytical methods to determine the absolute instead of the relative level of a large number of chemical compounds in biofluids. Routine analyses evaluate only a very restricted number of compounds, such as glucose level for monitoring diabetes, cholesterol and low/high-density lipoproteins for cardiovascular health, or urea and Creatinine for renal disorders. Simultaneously determining the absolute concentration of hundreds of molecules will open up new scenarios towards a more accurate personalized medicine and increase the predictive value of such analyses."

For example, when it comes to evaluating an existing SNR for a measurement to the detectability of a given peak (metabolite), here I am thinking of figure 1 and its consequences. The effect of the increase in SNR on the example of HC- is described here, after measuring the order of 4.5 min we have quite a decent R2. On the other hand, the R2 optimization procedure and its time consequences are not clearly explained.

We sincerely apologize to the reviewer, but we did not understand the relationship between "the order of 4.5 min" and R2. Regarding the R2 optimization procedure, we added a new sentence at line 209:

"This increase in the R2 value is a natural consequence of using the deconvolution process assisted by UHPLC-HRMS intensities, but its final value is still a measurement of the degree of agreement between these two datasets measured orthogonally. "

To sum up, the article is written quite decently, but some issues require clarification. I also believe that the purpose, the way to get there, and the actual possible applications (and the way to evaluate them) need to be outlined more clearly. This will undoubtedly increase the potential citation of the article. Adding a discussion about the possibility of using T1, T2 relaxometry or diffusometry, much cheaper techniques, may also contribute to the expansion of potential recipients of the article.

Although relaxation and diffusion-edited NMR spectra can undoubtedly add to the separation of different components in mixtures, they are not entirely quantitative, and their use in the SYNHMET context will be only of marginal importance.

Reviewer 3 Report

The authors present a strategy to obtain accurate concentrations of a high number of compounds in biofluids, like urine, to delineate a personalized metabolic profile, called: SYnergic use of NMR and HRMS for METabolomics (SYNHMET). They have reached 165 metabolites quantified in the urine of healthy subjects, patients with cystitis, and bladder cancer.

The paper is interesting and well written. The results have been achieved without the use of analytical standards and calibration curves and the obtained patient's personalized profile can have potential applications in clinical practice to monitor for example a disease progression.

I have only small considerations to improve:

  • In Figure 1 the letter’s size of the title and the numbers of both axes cannot be read well, as well as the equation of the regression line and the correlation coefficient. Please, enlarge them in the five graphics.
  • In Figure 4 the letter’s size of the “Biochemical Classification” names is too small to be read properly, as well as the titles “Control group”, “Chronic cystitis group” and “Bladder cancer group”.
  • In Figure 5 the letter’s size of the metabolites’ name is too small.

Typing errors/English spelling to check.

  • Line 19 delineate instead of de-lineate
  • Line 21 compared instead of compare
  • Line 90 help to achieve or help achieving instead of help achieve
  • Line 287 “,.” remove “,”

Finally, although there is a paragraph with the main conclusions of the work, there is no separate Conclusions section in the manuscript. It should be added as a separate heading.  

Author Response

The authors present a strategy to obtain accurate concentrations of a high number of compounds in biofluids, like urine, to delineate a personalized metabolic profile, called: SYnergic use of NMR and HRMS for METabolomics (SYNHMET). They have reached 165 metabolites quantified in the urine of healthy subjects, patients with cystitis, and bladder cancer.

The paper is interesting and well written. The results have been achieved without the use of analytical standards and calibration curves and the obtained patient's personalized profile can have potential applications in clinical practice to monitor for example a disease progression.

I have only small considerations to improve:

  • In Figure 1 the letter's size of the title and the numbers of both axes cannot be read well, as well as the equation of the regression line and the correlation coefficient. Please, enlarge them in the five graphics.

Character sizes of Figure 1 were increased.

  • In Figure 4 the letter's size of the "Biochemical Classification" names is too small to be read properly, as well as the titles "Control group", "Chronic cystitis group" and "Bladder cancer group".

Character sizes of Figure 4 were increased.

  • In Figure 5 the letter's size of the metabolites' name is too small.

Character sizes of Figure 5 were increased.

Typing errors/English spelling to check.

  • Line 19 delineate instead of de-lineate
  • Line 21 compared instead of compare
  • Line 90 help to achieve or help achieving instead of help achieve
  • Line 287 ",." remove ","

All typos have been corrected.

Finally, although there is a paragraph with the main conclusions of the work, there is no separate Conclusions section in the manuscript. It should be added as a separate heading.

A new section of conclusions has been added.  

Round 2

Reviewer 2 Report

I accept in present form.